# Radiofrequency Echographic Multi Spectrometry (R.E.M.S.): New Frontiers for Ultrasound Use in the Assessment of Bone Status—A Current Picture

**DOI:** 10.3390/diagnostics13101666

**Published:** 2023-05-09

**Authors:** Antonella Al Refaie, Leonardo Baldassini, Caterina Mondillo, Elisa Giglio, Michela De Vita, Maria Dea Tomai Pitinca, Stefano Gonnelli, Carla Caffarelli

**Affiliations:** Section of Internal Medicine, Department of Medicine, Surgery and Neuroscience, University of Siena, 53100 Siena, Italy

**Keywords:** Radiofrequency Echographic Multi Spectrometry, BMD, osteoporosis, REMS technology, fragility fractures, evaluation of bone status

## Abstract

Osteoporosis is a frequently occurring skeletal disease, and osteoporosis-related fractures represent a significant burden for healthcare systems. Dual-Energy X-ray Absorptiometry (DXA) is the most commonly used method for assessing bone mineral density (BMD). Today, particular attention is being directed towards new technologies, especially those that do not use radiation, for the early diagnosis of altered bone status. Radiofrequency Echographic Multi Spectrometry (REMS) is a non-ionizing technology that evaluates the bone status at axial skeletal sites by analyzing raw ultrasound signals. In this review, we evaluated the data on the REMS technique present in the literature. The literature data confirmed diagnostic concordance between BMD values obtained using DXA and REMS. Furthermore, REMS has adequate precision and repeatability characteristics, is able to predict the risk of fragility fractures, and may be able to overcome some of the limitations of DXA. In conclusion, REMS could become the method of choice for the assessment of bone status in children, in women of childbearing age or who are pregnant, and in several secondary osteoporosis conditions due to its good precision and replicability, its transportability, and the absence of ionizing radiation. Finally, REMS may allow qualitative and not just quantitative assessments of bone status.

## 1. Introduction

Osteoporosis is a skeletal disease that is characterized by low bone mass as well as microarchitectural alterations that increase bone fragility and risk of fractures [1]. Today, due to scientific progress and the aging population, the number of people suffering from chronic diseases has increased considerably, and the treatment of these conditions represents a significant burden for healthcare systems.

Therefore, for chronic diseases and for osteoporosis in particular, it is of crucial importance to focus attention on prevention and early diagnosis to reduce complications and the consequent socio-health costs. For these reasons, in recent years, there has been growing interest in identifying new parameters and technologies capable of diagnosing osteoporosis and predicting the risk of fragility fractures in a simple and economically sustainable way.

Among the imaging techniques available for bone assessment are the following:Dual-Energy X-ray Absorptiometry (DXA) measures bone mineral density (BMD) and is universally recognized as the gold standard for the diagnosis of osteoporosis [2]. Unfortunately, DXA has certain limitations; a 2015 study determined that 90% of DXA exams experienced errors (including errors while setting the location information for the patient, during patient positioning, and errors occurring during the analysis) [3]. Moreover, some artifacts can alter DXA results; for example, aortic calcification can lead to an overestimation of bone mineral density [4]; vertebroplasty, especially when involving two or more lumbar vertebrae, limits the ability to obtain an adequate assessment of BMD using DXA [5]; and osteoarthritis (OA), which represents the most common cause of artifacts when using DXA, especially in older patients [6]. In fact, the presence of structural abnormalities caused by OA (sclerosis, osteophytosis) artificially increases BMD measurements for the lumbar spine when using DXA [6].Quantitative ultrasonography (QUS) measures the transmission speed and attenuation of waves at the level of the heel, patella, and phalanges of the hands; however, QUS has seen little use because it does not measure bone status at axial skeletal sites and does not allow a diagnostic classification of osteoporosis.Quantitative Computed Tomography (QCT) accurately measures the relationship between cortical and trabecular bone portions; this exam represents the most specific method for assessing bone status but is also the most complex and is not always available. High-Resolution Peripheral Quantitative Computed Tomography (HR-pQCT) is used to assess bone density in the tibia and radius bone architecture; however, there is an unclear correlation between HR-pQCT and non-vertebral fractures and higher exposure to ionizing radiation, and cost do not allow this tool to be used in clinical practice [7,8].Magnetic Resonance Imaging (MRI) determines information about bone microarchitecture. It is a non-ionizing technique that is not currently usable in clinical practice because it is expensive, not readily available, and takes a long time to carry out.

Thus, imaging techniques for the evaluation of bone status have certain limitations or are expensive. It is precisely for this reason that there is growing interest in new non-invasive techniques that are able to assess bone status from both a quantitative and qualitative point of view.

In recent years, a new technique has captured the attention of global osteoporosis experts: REMS (Radiofrequency Echographic Multi Spectrometry).

REMS is a non-ionizing technology that evaluates bone status by analyzing raw, unfiltered native ultrasound signals, so-called radio frequency (RF) ultrasound signals, obtained during an ultrasound scan of the lumbar vertebrae and proximal femur. The analysis of native unfiltered ultrasound signals allows for information about the characteristics of bone tissue to be acquired. Bone density is obtained by comparing the spectra of analyzed signal with reference spectral models that have been previously obtained [9].

REMS scans are performed at both the proximal femur and lumbar spine using a 3.5 MHz convex ultrasound probe. The probe is placed on the hip and abdomen to visualize the interface of the target bone. The clinician regulates the depth and the focus of the transducer. The software detects the sought bone interfaces in the acquired frame sequence and identifies regions of interest for a diagnostic evaluation. The measured data are synthesized into a patient-specific spectrum that is compared to reference spectral models matched by gender, age, site, and BMI in a database. The spectral modifications introduced by the physical properties of bone structure that back-diffuse the ultrasound signals are identified via a comparison procedure to determine an estimate of the BMD and the consequent diagnostic classification of healthy, osteopenic, or osteoporotic [9,10].

In this paper, we want to create a picture of the current knowledge of this new and innovative technique in the literature Figure 1.

## 2. Materials and Methods

A literature review was conducted from inception (2019) to 31 January 2023. The Pubmed-Medline, Cochrane Library, ClinicalTrials.gov (accessed on 31 January 2023), and SCOPUS databases were searched using the following search terms: (“REMS densitometry” or “REMS technology” or “Radiofrequency Echographic MultiSpectrometry”) and (“osteoporosis” or “Bone Mineral Density” or “bone status”). The selection process for the studies included in the review is shown in Figure 2.

## 3. Results

### 3.1. Main Characteristics of the Studies on Validation of the REMS Technique

The first study on REMS validity was an Italian multicenter cross-sectional observational study by Di Paola et al. that compared the values of bone mineral density obtained by REMS to those obtained by DXA [11]. A total of 1914 postmenopausal women were evaluated. The results showed good agreement between the two methods, with a diagnostic concordance of 88.8% (k = 0.824, *p* < 0.001) for the lumbar spine (LS) and 88.2% (k = 0.794, *p* < 0.001) for the femoral neck (FN), respectively. Agreement between the two methods was also observed for the T-scores and was confirmed by linear regression [11]. Moreover, this study reported that the REMS technique presented very good precision and repeatability; in particular, precision was 0.38% (95% confidence interval: 0.28–0.48%) for the lumbar spine and 0.32% (0.24–0.40%) for the femoral neck.

These promising results were confirmed by Cortet et al. in a wider European population a few years later [12]. In this study, the diagnostic concordance between BMD determined using DXA and BMD determined using REMS was confirmed by considering a large population (4307 female Caucasian patients) with an older age range (from 30 to 90 years). In this study, both DXA and REMS had specificity and sensitivity ratings over 90% and a diagnostic concordance of about 86% for both the lumbar spine and proximal femur [12]. Moreover, the areas under the curve (AUCs) of the Receiver Operating Characteristic (ROC) curve evaluating the ability to discriminate groups of patients with previous osteoporotic fracture using DXA T-score and REMS T-score values were higher for REMS at both the femur and lumbar spine.

In conclusion, these two latter studies clearly demonstrated the diagnostic accuracy of REMS technology in the diagnosis of osteoporosis.

Later, the focus was on demonstrating the ability of REMS to predict the risk of fracture. A recent prospective observational study by Adami et al. carried out on 1370 women followed for 5 years evaluated the effectiveness of T-scores obtained by REMS to assess risk of fracture [13]. Data confirmed that the T-scores obtained by REMS were an effective predictor of fragility fractures, thus representing a further promising parameter for improving the diagnosis of osteoporosis in clinical practice [13].

The good correlation between diagnosing osteoporosis and predicting the risk of fracture by R.E.M.S. technology was confirmed in several other observational studies: one performed on 343 Brazilian women in 2021 [14]; one performed on a Polish population [15]; one performed on a Bulgarian population [16]; and, most recently, in two other 2022 studies (a study performed on 455 Mexican women [17] and an Italian study performed on 175 patients) [18].

Moreover, a recent prospective observational study by Pisani et al. evaluated the usefulness of an additional REMS parameter, the Fragility Score (FS), in the prediction of incident fragility fractures. FS is a dimensionless parameter that is calculated by comparing the patient-specific spectra to reference models for “frail” and “non-frail” bone spectra that have been previously obtained from subjects with or without fragility fractures. Therefore, FS investigates characteristics related to bone quality and microarchitecture, thus providing a fracture risk estimation independently of BMD [19]. Previously, Greco et al. found a good correlation between the Fragility Score and FRAX^®^ for the fracture risk estimation (r = 0.71; *p* < 0.001) in a female population [20]. The study by Pisani et al. carried out on a cohort of 1989 Caucasian patients of both genders who were from 30 to 90 years of age, evaluated the incidence of fractures during a follow-up period of up to 5 years; this study reported adequate intra-operator and inter-operator repeatability and demonstrated the predictive capacity of the FS to identify patients who were at risk of incident fragility fractures (AUC = 0.811 for women and AUC = 0.780 for men). Moreover, the ability of FS to identify subjects who were at risk for fragility fractures was better with respect to that of REMS when using the BMD and DXA BMD T-scores [19]. This last study is important because it also confirmed the diagnostic usefulness of the REMS technique in a large male population. The main characteristics of the studies on the validation of the REMS technique are reported in Table 1.

### 3.2. Main Characteristics of the Studies on the Use of REMS Technology in Real-Life Clinical Practice

Despite osteoporosis diagnosis being based on BMD values, the risk of fracture does not only depend on BMD: bone quality, bone strength, and many other factors, such as cellular density, bone mineralization, and trabecular and cortical properties, including thickness, porosity, and bone microarchitecture, are fundamental [7].

The potential capability of REMS to evaluate parameters related to bone quality and strength could be particularly useful for assessment in subjects with secondary osteoporosis. In fact, in secondary osteoporosis, BMD may be normal or only slightly reduced, but the risk of fragility fracture may still be increased. For this reason, various studies on REMS have aimed to evaluate bone status in these diseases. The most typical example of the usefulness of REMS in this field is represented by “diabetic osteopathy”. In particular, diabetes mellitus type 2 (T2DM), the most frequent metabolic disease in the world, is characterized by a higher risk of fragility fractures with respect to non-diabetic subjects despite similar or higher BMD levels, suggesting that in T2DM, qualitative bone alterations may play an important role in bone fragility. A 2020 study by Caffarelli et al. compared DXA and REMS in assessing the bone status in 90 women affected by T2DM and in 90 age-matched healthy controls [21]. All patients underwent DXA and REMS exams. The DXA measurements were all higher in the T2DM women than they were in the non-T2DM women; instead, the REMS measurements were lower in the T2DM women than they were in the controls at both the lumbar spine and proximal femur. Moreover, the percentage of T2DM women classified as “osteoporotic” on the basis of BMD by REMS was markedly higher with respect to those classified by DXA (47.0% vs. 28.0%, respectively). In addition, the percentage of T2DM women classified as osteopenic or normal by DXA was higher with respect to those classified by REMS (48.8% and 23.2% vs. 38.6% and 14.5%, respectively). The T2DM women with previous major fragility fractures presented lower values of both BMD-LS obtained using DXA and BMD-LS measured obtained using REMS with respect to those without fractures; the difference was only significant for BMD-LS obtained using REMS. In conclusion, data from this study suggest that REMS technology may be more sensitive when assessing bone status in T2DM patients than DXA and may represent a useful approach to enhance the diagnosis of diabetic osteoporosis and reduce fragility fractures [21].

Moreover, a Polish study conducted in 2020 evaluated bone status in a group of patients with acromegaly using REMS technology and found that BMD was reduced at both the lumbar spine and proximal femur [22].

In 2019, Bojincă et al. carried out a cross-sectional observational study on another frequently diagnosed type of secondary osteoporosis: rheumatoid arthritis (RA), using REMS. This study reported that RA patients presented lower values of BMD with respect to the controls at all skeletal sites and that RA patients had a higher fracture risk and higher prevalence of osteoporosis [23].

Another strength that emerged from data present in the literature is that the REMS technique may be able to overcome the most common artifacts, including osteoarthritis, vascular calcifications, and vertebroplasty of the lumbar spine, which affect the BMD value obtained by DXA. This possibility was first demonstrated in a paper reporting several case reports [24] and that has since become the starting point for other extended studies. Until a few years ago, osteoporosis and osteoarthritis (OA), two common chronic diseases linked to aging, were considered two completely independent nosological entities. Recently, however, it has been demonstrated that osteoporosis and OA are both often present in patients and are linked by complex physiopathological mechanisms [6,25]. Some studies have also reported that women with OA present alterations in bone structure and have an increased risk of fractures [25,26]. In particular, the characteristic alterations of OA, such as osteophytosis, bone sclerosis, and arthrosis of the facet joints, are particularly present at the level of the lumbar spine and can artificially increase the BMD levels measured by DXA.

In a recent study by Caffarelli et al. carried out on a cohort of 180 Caucasian women (66.2 ± 11.6 years) with radiologically diagnosed osteoarthritis, all subjects underwent both DXA evaluation and an echographic scan using REMS [27]. The mean BMD values at different skeletal sites obtained using the DXA and REMS techniques showed that the BMD T-scores obtained using REMS were significantly lower than those obtained using DXA both at the lumbar spine (*p* < 0.01) and at all femoral subregions (*p* < 0.05). In OA subjects, the percentage of women classified as “osteoporotic” on the basis of BMD according to REMS was markedly higher with respect to those classified using DXA (35.1% vs. 9.3%, respectively). Similarly, REMS allowed a greater number of patients with fractures to be classified as osteoporotic than DXA (58.7% vs. 23.3%, respectively). In conclusion, REMS appears to be able to overcome common artifacts such as structural alterations caused by OA at the lumbar spine, which affect the BMD values obtained by DXA [27]. Chronic kidney diseases (CKDs) are associated with mineral and bone diseases, including pain, bone loss, and fragility fractures [28]. Moreover, when CKD patients undergo hemodialysis and peritoneal dialysis, the risk of fragility fracture markedly increases [29]. Unawareness of such complications has led to poor fracture management and a lack of preventive approaches. Unfortunately, in these patients, the evaluation of BMD by DXA presents some important limits; in fact, the presence of aortic and ligamentous calcifications (very frequent in CKD patients), degenerative changes, and scoliosis lead to overestimates of BMD by DXA. Moreover, patients with severe CKD are often debilitated and cannot undergo DXA. A recent work by Fassio et al. [30] aimed to verify whether the REMS technique could actually play a role in bone assessment for CKD patients undergoing peritoneal dialysis. Forty-one patients undergoing peritoneal dialysis were enrolled in the study. All patients underwent DXA and REMS exams to evaluate bone status. The analysis concluded that there is good agreement between the two techniques for BMD evaluation at the femur. At the lumbar spine, the DXA anteroposterior mean T-score (−0.49 ± 1.98) was significantly higher than both the laterolateral DXA (−1.66 ± 0.99) and the REMS (−2.00 ± 1.94) measurements (*p* < 0.01 for both). The discrepancy between the DXA anteroposterior lumbar T-score and DXA laterolateral and REMS T-score was positively associated with the extent and severity of aortic calcifications. No statistically significant differences in the DeFRA^®^ and FRAX^®^ outputs were found when calculated using the DXA and REMS data. Therefore, the REMS technique, which is transportable and can be moved to the patient’s bedside and is able to exclude calcifications, should be able to overcome DXA limits in CKD patients undergoing peritoneal dialysis [30].

The crucial challenge for REMS is the possibility to assess bone status and risk of fracture without the use of ionizing radiation in young, fertile populations with fracture risk, such as subjects with metabolic bone diseases, malabsorption, osteogenesis imperfecta, chronic inflammatory diseases, rheumatologic diseases treated with corticosteroids, and patients treated with drugs associated with osteoporosis. The first study that considered a young population was an Italian study by Caffarelli et al. that compared BMD values obtained using DXA and REMS in a population of 50 young women suffering from anorexia nervosa (AN) [31]. The study confirmed the good precision of REMS technology in the assessment of bone status in a young population. Moreover, in this study, the BMD values obtained using DXA and the BMD values obtained using REMS expressed as Z-scores were all significantly lower in AN patients than in the controls. The subjects suffering from AN with previous vertebral fragility fractures presented lower BMD values at both the lumbar spine and total hip measured when measured using DXA and REMS with respect to those without fractures; however, the difference was significant only for BMD-total hip when obtained by REMS. Moreover, Bland–Altman plots confirmed that there was good agreement between the two techniques. Therefore, REMS technique could represent the most suitable tool for monitoring bone status in young women with AN [31].

The possibility of evaluating bone status by ultrasound in young women is revolutionary because ultrasound can be used in a safe way when subjects are of fertile age and also during pregnancy and when breastfeeding [31]. Bone is a dynamic tissue with constant turnover, and this is especially the case in women: hormonal changes, pregnancy, breastfeeding, and menopause have a particular influence on the skeleton. During pregnancy, bone represents the most important source of calcium for the fetus, and, in addition, reduced physical activity and less sun exposure can predispose one to a reduction in BMD. This pattern was demonstrated using REMS technology in an interesting paper that evaluated BMD using REMS in a prospective case–control observational study that compared BMD values obtained using REMS at the proximal femur in 78 pregnant women at 39 ± 1.5 weeks compared to non-pregnant women of the same age [32]. To our knowledge, this is the first study carried out by REMS technique that has demonstrated how BMD in pregnant women was lower (about −8.1%) compared to the controls. Moreover, femoral neck BMD presented a positive correlation with the pre-pregnant BMI and a negative correlation with maternal age [32]. These results confirm data present in the literature in which BMD was evaluated using DXA [33]. Thus, REMS evaluation is an attractive solution because it would allow clinicians to identify women with reduced BMD and who are at a greater risk of fractures so that they could be followed, even during pregnancy. The main characteristics of the studies on the use of REMS technology in real-life clinical situations are reported in Table 2.

### 3.3. Future Perspectives

The studies that we have examined in this review clearly indicate how the REMS technique has numerous advantages and great development opportunities in many areas in the field of bone diseases. In particular, due to its precision and repeatability, REMS technology has great potential for therapeutic monitoring and follow-up.

Furthermore, there are preliminary studies about the use of REMS on special populations, such as in those with rare bone diseases and with immobility conditions. The REMS device is easily transportable, and measurements can be carried out directly at the patient’s bedside (such as in patients with a recent hip fracture). Due to the lack of ionizing radiation, REMS may become the preferable technique for assessing BMD in childhood. Some preliminary studies have explored the accuracy of REMS technology for bone assessment in pediatric-aged and adolescent patients. These points are being explored, and in the future, they could provide great advantages in clinical practice.

Another attractive future perspective is the possibility of evaluating muscle using the REMS technique. Today, it is well known that bone and muscle have a mutual interaction during formation, repair, and regeneration. Nowadays, musculoskeletal diseases (such as sarcopenia and osteoporosis) often co-exist, especially in individuals with chronic and metabolic diseases (COPD, diabetes, chronic inflammatory diseases, obesity); in fact, systemic inflammation negatively affects bone quality and muscle health, strength, and function. A new perspective in this field regarding REMS technology that could be used not only for bone health evaluation but also, if using a new software, to obtain information related to musculoskeletal health. Therefore, using a single ultrasound exam to provide complete musculoskeletal evaluation and appropriate follow-up would be a novel innovation.

## 4. Conclusions

To sum up, REMS technology has proven its worth in osteoporosis diagnosis and in predicting fracture risk in large populations. After reviewing the current works in the literature, it was determined that there are several advantages of this technology, such as 1. the possibility to evaluate bone quality; 2. the ability to overcome some of the limitations of DXA; 3. the possibility to allow clinicians to assess bone status with periodic follow-up without radiation when there are no other usable radiological methods (e.g., childhood, pregnancy); and 4. the transportability, ease of use, and economic sustainability of REMS.

Future works and ongoing research will help us understand the role of REMS in diagnosis and follow-up in osteoporosis patients with respect to DXA and will also help us understand target populations in which REMS could represent a better alternative to DXA in assessing bone quality and fracture risk.

## Figures and Tables

**Figure 1 diagnostics-13-01666-f001:**
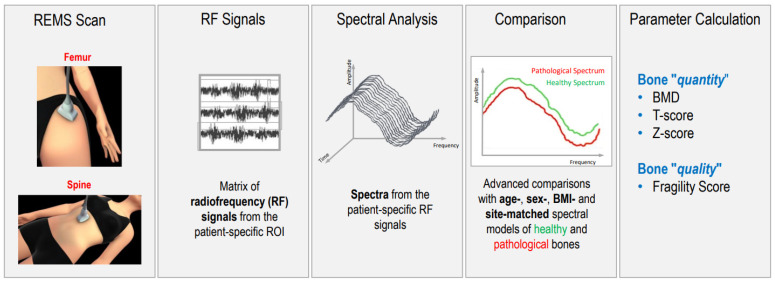
Schematic representation of radiofrequency echographic multispectrometry (REMS).

**Figure 2 diagnostics-13-01666-f002:**
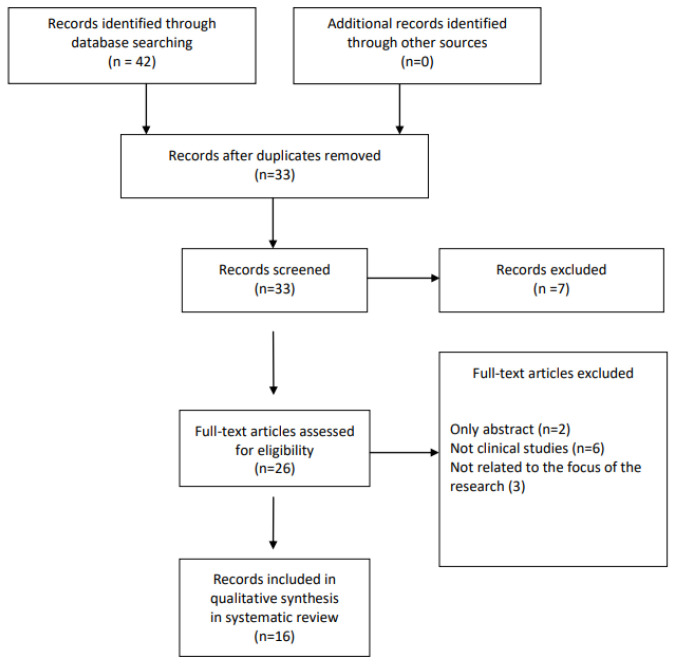
Flow chart of the studies identified and included in the review.

**Table 1 diagnostics-13-01666-t001:** Main characteristics of the studies on validation of the REMS technique.

Study/Year	Study Characteristics	Population	Outcomes	Results	Conclusions
Di Paola M (2019) Italy [11]	Multicenter cross-sectional observational	1914 PMO ♀ (51–70 years)	DXA-LS; DXA-FN; REMS-LS; REMS-FN	Sensitivity DXA vs. REMS LS = 91.7% FN = 91.5% Specificity DXA vs. REMS LS = 92.0% FN = 91.8% Diagnostic Concordance DXA vs. REMS LS = 88.8% (k = 0.824, *p* < 0.001) FN = 88.2% (k = 0.794, *p* < 0.001) Correlation DXA vs. REMS LS = (r = 0.94, *p* < 0.001) FN = (r = 0.93, *p* < 0.001)	REMS approach had a good level of accuracy and precision in osteoporosis diagnosis
Kirilova E (2019) Bulgaria [16]	Cross-sectional observational	25 premenopausal ♀: (24–50 years) 140 PMO ♀ (38–86 years)	REMS-LS; REMS-FN; REMS-TH	REMS-LS, REMS-FN, REMS-TH measurements of PMO group were significantly lower than those of premenopausal OP	REMS identify lower BMD between PMO and premenopausal OP
Adami G. (2020) Italy [13]	Longitudinal observational (5 years)	1370 ♀ (30–90 years)	DXA-LS; DXA-FN; REMS-LS; REMS-FN; Incidence of fragility fractures	Fracture incidence was 14.0% For a T-score cut-off −2.5 identified Fx REMS-LS = sensitivity of 65.1% and specificity of 57.7% DXA-LS = sensitivity of 57.1% and a specificity of 56.3% REMS-FN = sensitivity 40.2% and specificity 79.9% DXA-FN = sensitivity 42.3% and specificity 79.3%.	REMS T-score resulted an effective predictor of fragility fractures
Cortet B (2021) Europe [12]	Multicenter cross-sectional observational	4307 ♀ (30–90 years)	DXA-LS; DXA-FN; REMS-LS; REMS-FN;	Sensitivity DXA vs. REMS LS = 90.9% FN = 90.4% Specificity DXA vs. REMS LS = 95.1% FN = 95.5% ability to recognize fractured patients DXA and REMS The AUCs of the ROC curve: LS = 0.603 and 0.640 (*p* = 0.0002) FN = 0.631 for DXA and 0.683 for REMS (*p* < 0.001).	Diagnostic effectiveness of REMS technology was reconfirmed in a larger and younger population
Nowakowska-Płaza A (2021) Poland [15]	Cross-sectional observational	98 ♀ and 18 ♂ (40–90 years)	DXA-LS; DXA-FN; REMS-LS; REMS-FN;	diagnostic agreement DXA vs. REMS LS = 82.8% FN = 84.8%	Significant diagnostic agreement between DXA and REMS
Amorim DMR (2021) Brazil [14]	Cross-sectional observational	343 ♀ (30–80 years)	DXA-LS; DXA-FN; REMS-LS; REMS-FN;	Correlation DXA vs. REMS LS = (r = 0.75, *p* < 0.001) FN = (r = 0.78, *p* < 0.001) The AUCs of the ROC curve using DXA T-score as reference: LS = 0.94; FN = 0.97	REMS comparing with DXA had high accuracy for the diagnosis of osteoporosis
Sergio RO (2022) Mexico [17]	Cross-sectional observational	455 ♀ (40–87 years)	REMS-LS; REMS-FN;	diagnostic agreement between REMS-LS and REMS-FN = 73%. Good correlation between LS, FN by REMS Prevalence of OP ↑ with age and ↓ BMI	Age and BMI variations correlate with the prevalence of osteoporosis.
Lalli P (2022) Italy [18]	Cross-sectional observational	140 primary OP: 120 ♀, 20 ♂ (64–81 years) 35 disuse-related OP: 14 ♀; 20 ♂ (49–65.3 years)	DXA-LS; DXA-FN; REMS-LS; REMS-FN;	Diagnostic concordance DXA and REMS Primary OP = 63% (Cohen’s kappa = 0.31) Disuse-relate OP = 13% (Cohen’s kappa = −0.04) Correlation FS and FRAX major Fx = (R = 0.65, *p* = 0.0001) Correlation FS and hip Fx (R = 0.62, *p* = 0.0001) in primary OP no Fx	REMS showed excellent test–retest reproducibility diagnostic concordance between DXA and REMS was minimal
Pisani P (2023) Italy [19]	Longitudinal observational (5 years)	1289 ♀ (54–66 years) Fx: 181 (63–74 years) No Fx: 1108 (54–64 years) 515 ♂ (48.3–73.0 years) Fx: 67 (57.3–78 years) No Fx: 448 (47–71 years)	DXA-LS; DXA-FN; REMS-LS; REMS-FN; FS Incident Fx	For prediction of generic fracture FS provided AUC = 0.811 for ♀ and AUC = 0.780 for ♂, which resulted in AUC = 0.715 and AUC = 0.758 adjusted for age and BMI For prediction of hip fracture, the corresponding values were AUC = 0.780 for ♀ and AUC = 0.809 for ♂, which became AUC = 0.735 and AUC = 0.758 adjusted for age and BMI	FS displayed a superior performance in fracture prediction, representing a valuable diagnostic tool to accurately detect a short-term fracture risk

Abbreviations: PMO: postmenopausal osteoporosis; ♀: female; DXA: Dual Energy X-ray Absorptiometry; LS: Lumbar Spine; FN: Femoral Neck; REMS: Radiofrequency Ecographic Multi Spectrometry; OP: osteoporosis; TH: Total Hip; ↓: Reduction; ↑: Increase; Fx: Fracture; AUCs: Area Under the Curve; ♂: male; FS: Fragility score; and BMI: Body Mass Index.

**Table 2 diagnostics-13-01666-t002:** Main characteristics of the studies on the use of REMS technology in real-life clinical practice.

Author/Year, Country	Study	Population	Assessments	Results	Conclusions
Bojincă VC (2019) Romania [23]	Cross-sectional observational	RA: 106 ♀ (65 ± 8 years) controls: 119 ♀ (64 ± 13 years)	REMS-LS; REMS-FN dx; REMS-FN sn	RA patients ↓ BMD at all sites RA had higher prevalence of osteoporosis	REMS is able to replicate the results of the established DXA measurements
Rolla M (2020) Poland [22]	Cross-sectional observational	AG: 25 ♀, 8 ♂ (59.1 ± 9.8 years) CG: 17 ♀, 7 ♂ (age-matched)	DXA-LS; DXA-FN; REMS-LS; REMS-FN	REMS BMD-LS, T-score LS and Z-score LS and BMD-FN, T-score FN, Z-score FN are in agreement with DXA measurements in AG and CG.	REMS may be considered a potential method in assessment of bone status in acromegaly
Caffarelli C (2021) Italy [21]	Cross-sectional observational	TDM2: 90 ♀ (70.5 ± 7.6 years) controls: 90 ♀ (69.2 ± 7.5 years)	DXA-LS; DXA-FN; DXA-TH REMS-LS; REMS-FN; REMS-TH	REMS BMD ↓ T2DM than in non T2DM REMS classified as “osteoporotic” more T2DM respect to those classified by DXA (47.0% vs. 28.0%, respectively)	REMS technology may represent a useful approach to enhance the diagnosis of osteoporosis in patients with T2DM
Degennaro VA (2021) Italy [32]	Cross-sectional case—control observational	pregnant: 78 ♀ (32.9 ± 5.0 years) controls: 78 ♀ (32.9 ± 5.2 years)	REMS-FN	REMS-FN BMD ↓ (8.1%) in pregnant women than in controls	It is the first study that demonstrate decreased BMD in pregnancy thanks to REMS
Caffarelli C (2022) Italy [27]	Cross-sectional observational	OA: 113 ♀ (63.2 ± 11.3 years) Vertebral Fx: 43 ♀ (73.6 ± 18.5 years)	DXA-LS; DXA-FN; DXA-TH REMS-LS; REMS-FN; REMS-TH	REMS BMD T-scores LS, FN and TH were significantly lower than DXA BMD T-score LS (*p* < 0.01) FN and TH (*p* < 0.05). In OA group REMS classified as “osteoporotic” more subjects respect to those classified by DXA (35.1% vs. 9.3%, respectively). In Vertebral Fx group REMS classified as “osteoporotic” more subjects respect to those classified by DXA (58.7% vs. 23.3%, respectively).	REMS appears to be able to overcome the most common artifacts, such as OA and vertebral Fxe of the lumbar spine, which affect the value of BMD by DXA.
Caffarelli C (2022) Italy [31]	Cross-sectional observational	AN: 47 ♀ (31.7 ± 10.3 years) controls: 30 ♀ (32.9 ± 9.5 years)	DXA-LS; DXA-FN; DXA-TH REMS-LS; REMS-FN; REMS-TH	Correlation DXA vs. REMS LS = (r = 0.64, *p* < 0.01) FN = (r = 0.86, *p* < 0.01) TH = (r = 0.84, *p* < 0.01) Good agreement REMS between DXA by Bland–Altman analysis AN with Fx have lower values of both BMD-LS and BMD-TH by DXA and by REMS with respect to AN without Fx	REMS represent an important tool for the evaluation AN in young women, especially during the fertile age and in case of pregnancy and breastfeeding.
Fassio A (2023) Italy [30]	Cross-sectional observational	41 (29♂; 12♀) (61.1 ± 13.7 years)	DXA-LS-AP; DXA-LS-LL; DXA-FN; DXA-TH REMS-LS; REMS-FN; REMS-TH	No significant differences between BMD T-scores FN and TH measured by DXA or REMS BMD-LS-AP by DXA was higher (−0.49 ± 1.98) respect to BMD-LS-LL (−1.66 ± 0.99) by DXA and BMD-LS (−2.00 ± 1.94) by REMS	promising agreement, in a real-life PD setting, between the DXA and REMS BMD values and in the consequent fracture risk assessment.

Abbreviations: RA: rheumatoid arthritis; **♀**: female; DXA: Dual Energy X-ray Absorptiometry; LS: Lumbar Spine; FN: Femoral Neck; REMS: Radiofrequency Ecographic Multi Spectrometry; BMD: Bone Mineral density; ↓: Reduction; OP: osteoporosis; ♂: male; AG: patients with acromegaly; CG: control group; T2DM: Type 2 Diabetes Mellitus; TH: Total Hip; OA: Osteoarthritis; Fx: Fracture; AN: anorexia nervosa; PD: peritoneal dialysis; AP anteroposterior; and LL: latero-lateral scan.

## Data Availability

Not applicable.

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
