# Peer review of "Radiofrequency Echographic Multi Spectrometry (R.E.M.S.): New Frontiers for Ultrasound Use in the Assessment of Bone Status—A Current Picture"

_diagnostics, 2023, doi:10.3390/diagnostics13101666_

Round 1

Reviewer 1 Report

A review manuscript (ID: diagnostics-2282295) entitled: "Radiofrequency Ecographic MultiSpectrometry (R.E.M.S.): new ultrasound frontier in assessment of bone status, a current picture."has been  submitted by Antonella Al Refaie, Leonardo Baldassini, Caterina Mondillo, Elisa Giglio, Michela De Vita, Maria Dea Tomai Pitinca, Stefano Gonnelli and Carla Caffarelli to the section: Medical Imaging and Theranostics of the scientific journal Diagnostics. The review is well written and can be be considered as a valuable contribution to this field. Nevertheless, the authors should carefully check if all relevant publications are cired. For example I have missed a discussion of the following publication:

Cezary Iwaszkiewicz, Piotr Leszczyński (2019) Bone densitometry by radiofrequency echographic multi-spectrometry (REMS) in the diagnosis of osteoporosis. Forum Reumatologiczne 5(2): 81-88. DOI: 10.5603/FR.2019.0011

Author Response

Reviewer 1

A review manuscript (ID: diagnostics-2282295) entitled: "Radiofrequency Ecographic MultiSpectrometry (R.E.M.S.): new ultrasound frontier in assessment of bone status, a current picture."has been  submitted by Antonella Al Refaie, Leonardo Baldassini, Caterina Mondillo, Elisa Giglio, Michela De Vita, Maria Dea Tomai Pitinca, Stefano Gonnelli and Carla Caffarelli to the section: Medical Imaging and Theranostics of the scientific journal Diagnostics. The review is well written and can be be considered as a valuable contribution to this field. Nevertheless, the authors should carefully check if all relevant publications are cired. For example I have missed a discussion of the following publication:

Cezary Iwaszkiewicz, Piotr Leszczyński (2019) Bone densitometry by radiofrequency echographic multi-spectrometry (REMS) in the diagnosis of osteoporosis. Forum Reumatologiczne 5(2): 81-88. DOI: 10.5603/FR.2019.0011

According to the suggestion of the Referee, the publication has been added in the “References” section [10]. 

Reviewer 2 Report

The authors would take a picture of the current knowledge in literature about 88 this new and innovative technique. The conclusion of this systematic review is REMS technology has proven its worth in osteoporosis diagnosis and in predicting fracture risk in large populations. So reviewing the works currently in the literature, the advantages of this technology are several:1.the possibility to evaluate bone quality; 2. the ability to overcome some of DXA limitations; 3.the possibility to allow clinicians to assess bone status with periodic follow-up without radiation, when there are no other usable radiological methods (e.g. childhood, pregnancy); 4. REMS is also  transportable, easy to use and economically sustainable. 

The introduction is well written , with adequate bibliographic references . 

The methodology is complete, widely described, which would allow the study to be carried out by another research group. The description of the results is confusing and the data is presented in a motley manner, which makes it difficult to read. The authors' description of the possibility of measuring bone quality parameters is interesting. This has been indicated for the QUS previously The conclusion are clearly described  

Author Response

Thank you for your comments and corrections.

The authors would take a picture of the current knowledge in literature about this new and innovative technique. The conclusion of this systematic review is REMS technology has proven its worth in osteoporosis diagnosis and in predicting fracture risk in large populations. So reviewing the works currently in the literature, the advantages of this technology are several:1.the possibility to evaluate bone quality; 2. the ability to overcome some of DXA limitations; 3.the possibility to allow clinicians to assess bone status with periodic follow-up without radiation, when there are no other usable radiological methods (e.g. childhood, pregnancy); 4. REMS is also  transportable, easy to use and economically sustainable.

The introduction is well written , with adequate bibliographic references .

The methodology is complete, widely described, which would allow the study to be carried out by another research group. The description of the results is confusing and the data is presented in a motley manner, which makes it difficult to read. The authors' description of the possibility of measuring bone quality parameters is interesting. This has been indicated for the QUS previously The conclusion are clearly described 

Thank you for your comments and corrections. Extensive editing of English language and style was performed, therefore we believe that the understanding of the results section is clearer.

Reviewer 3 Report

This paper reviews advantages of Radiofrequency Ecographic MultiSpectrometry (REMS) for diagnosis of bone mineral density disorders. The paper's presentation is inadequate for publishing and needs extensive editing for English language problems. Please see attached pdf file for additional comments and corrections.

Author Response

According to the suggestion of the Referee we submitted the manuscript to a extensive editing of English language and style

According to the suggestion of the Referee, in Figure 1 we change lumbar spine with spine.

According to the suggestion of the Referee, we change inception with 2019

At present, preliminary data subject to abstracts at regional congresses are available

Abstract on  XXII Congresso Nazionale SIOMMMS, Bari 13-15 October 2022:  Paola Pisani, Alessandra Natale, Fiorella Anna Lombardi, Tommaso De Marco, Luigi Antelmi, Alessia Centonze, Francesco Conversano, Sergio Casciaro. “REMS: un approccio dedicato per la valutazione dello stato di salute muscolare all’avambraccio”

Reviewer 4 Report

This work explores the use of REMS in clinical practice for the diagnosis of osteoporosis along with applications in other conditions such as RA, pregnancy etc using data extracted for a relative search in the literature. The authors report that REMS findings are in most of the cases in agreement with the gold standard DEXA measurements and conclude the REMS could be useful in assessing bone quality and could overcome other imaging techniques’ limitations.

The topic is interesting and the authors provide an up-to-date summary of the existing data. However, they need to address some points.

·         The manuscript has a number of spelling and grammar errors and, thus, needs a very careful editing. For example, line 15 - “at axial”, line 20 – “in conclusion”, line 31 – “thanks to advances in medicine” must be deleted or rephrased, line 43 – “Even DXA is unfortunately without limits” must be rephrased, line 65 – “because expensive”, line 251 – “corticosteroidesor” etc. This can be observed throughout the text.

·         Line 44, are errors in geographical information due to DEXA application? This is not clear.

·         The authors state that QUS not allow a diagnostic classification of osteoporosis. Is there any evidence for this? Other articles claim that calcaneus QUS could be applied (e.g. PMID: 34344965)

·         The same stands for pQCT. Citations are needed to support that certain conditions “do not allow to use this tool in clinical practice”.

·         Line 67, which of the aforementioned imaging techniques are invasive?

·         A major issue in this manuscript is the nature of the work and the use of PRISMA. Do the authors aim to perform a systematic review or write a summary of the existing data? PRISMA is traditionally used for systematic reviews and has been recently updated (the 2009 flowchart is out of date). If the authors want to conduct a systematic review, a risk of bias assessment for all included articles is essential. In addition, since all data for diagnostic accuracy have been extracted (e.g. AUCs), a meta-analysis would be also useful. This needs to be clarified.

·         The flowchart needs a better presentation and alignment. What does “Full-text articles excluded, with reasons (n=1)” mean?

·         Line 132, R.E.M.S. – consistency in abbreviations is essential (AN or ANX?).

·         Line 267 – is [29] the correct citation for this sentence?

·         Lines 275-7 – other works have also assessed BMD during pregnancy, e.g. PMID: 21607805, PMID: 8636299. Please comment.

Author Response

  • The manuscript has a number of spelling and grammar errors and, thus, needs a very careful editing. For example, line 15 - “at axial”, line 20 – “in conclusion”, line 31 – “thanks to advances in medicine” must be deleted or rephrased, line 43 – “Even DXA is unfortunately without limits” must be rephrased, line 65 – “because expensive”, line 251 – “corticosteroidesor” etc. This can be observed throughout the text.

According to the suggestion of the Referee we submitted the manuscript to a extensive editing of English language and style

  • Line 44, are errors in geographical information due to DEXA application? This is not clear.

According to the suggestion of the Referee we have remove the sentence in geographical information

  • The authors state that QUS not allow a diagnostic classification of osteoporosis. Is there any evidence for this? Other articles claim that calcaneus QUS could be applied (e.g. PMID: 34344965)

QUS does not allow a classification according to WHO criteria. Moreover, in some conditions, results obtained by QUS are not always concordant. As in diabetic patients for example in which data about the predictive role of QUS in discriminating diabetic patients with fragility fractures are conflicting. Furthermore the fact that QUS measurements can only be performed at non-axial skeletal sites and the availability of many devices that differ from each other in technology and measured parameters limits the clinical use of QUS.

  • The same stands for pQCT. Citations are needed to support that certain conditions “do not allow to use this tool in clinical practice”.

We added two publications that support this statement [7,8]

  • Line 67, which of the aforementioned imaging techniques are invasive?

According to the suggestion of the Referee we have removed the term “invasive”

  • A major issue in this manuscript is the nature of the work and the use of PRISMA. Do the authors aim to perform a systematic review or write a summary of the existing data? PRISMA is traditionally used for systematic reviews and has been recently updated (the 2009 flowchart is out of date). If the authors want to conduct a systematic review, a risk of bias assessment for all included articles is essential. In addition, since all data for diagnostic accuracy have been extracted (e.g. AUCs), a meta-analysis would be also useful. This needs to be clarified.

According to the suggestion of the Referee the Authors perform a summary of the existing data. We remove PRISMA form and added Fig 2: “Flow chart of the studies identified and included in the study”

  • The flowchart needs a better presentation and alignment. What does “Full-text articles excluded, with reasons (n=1)” mean?

According to the suggestion of the Referee the Authors we have removed the article with reason

  • Line 132, R.E.M.S. – consistency in abbreviations is essential (AN or ANX?).

According to the suggestion of the Referee we change abbreviations ANX in AN

  • Line 267 – is [29] the correct citation for this sentence?

According to the suggestion of the Referee we checked and corrected the references

  • Lines 275-7 – other works have also assessed BMD during pregnancy, e.g. PMID: 21607805, PMID: 8636299. Please comment.

According to the suggestion of the Referee we have added a sentence on BMD evaluation by DXA in pregnant women  and added a reference [33]

Round 2

Reviewer 2 Report

The results are more clearly described

Reviewer 4 Report

The authors addressed my comments sufficiently. Therefore, the revised manuscript can be accepted for publication.